# Web of Things Platforms for Distance Learning Scenarios in Computer Science Disciplines: A Practical Approach

**Llanos Tobarra**[ID]**, Antonio Robles-Gómez**[ID]**, Rafael Pastor** *[ID]**, Roberto Hernández, Jesús Cano**[ID] **and Daniel López**

Communications and Control Systems Department, Computer Science Faculty, Spanish National University for Distance Education (UNED), 28040 Madrid, Spain; llanos@scc.uned.es (L.T.); arobles@scc.uned.es (A.R.-G.); roberto@scc.uned.es (R.H.); jcano@scc.uned.es (J.C.); daniel.lopez@pas.uned.es (D.L.)

*   Correspondence: rpastor@scc.uned.es

**Abstract:** Problem-based learning is a widely used learning methodology in the field of technological disciplines, especially in distance education environments. In these environments, the most used tools, which provide learning scenarios, are remote and virtual laboratories. Internet of Things (IoT) devices can be used as remote or virtual laboratories. In addition to this, they can be organized/orchestrated to build remote maker spaces through the web. These types of spaces are called the Web of Things (WoT). This paper proposes the use of these types of spaces and their integration as practical activities into the curricula of technological subjects. This approach will allow us to achieve two fundamental objectives: (1) To improve the academic results (grades) of students; and (2) to increase engagement and interest of students in the studied technologies, including IoT devices. These platforms are modeled using archetypes based on different typologies and usage scenarios. In particular, these usage scenarios will implement a learning strategy for each problem to be solved. The current work shows the evolution of these archetypes and their application in the teaching of disciplines/subjects defined in computer science, such as distributed computing and cybersecurity.

**Keywords:** Internet of Things (IoT); Web of Things (WoT); computer science education; cloud computing; learning analytics

## 1. Introduction

The internet is now a big and complex example of a globally scalable network, composed of elements such as computers and devices. These elements interact among themselves across heterogeneous software and hardware platforms by using open and simple standards, which have enabled flexible, good performance and the building of scalable systems. Additionally, recent developments in the field of smart embedded devices have allowed users to have a wide variety of physical objects (or "things") integrated into smaller and smaller units, to capture and control the environment by wireless communications.

As a consequence, paradigms such as Pervasive Computing [1] and the Internet of Things (IoT) [2] have taken advantage, and have rapidly gained ground in both professional and domestic fields, by means of the pervasive presence of these embedded devices that interoperate among them to share services and information [3,4]. However, due to this massive deployment of everyday objects, the security risks and threats could reach a global dimension.

On the other hand, the integration of these low-cost and small devices in an application architecture using web technologies is known as the Web of Things (WoT) [5–7]. Thus, WoT provides

an application layer which allows things to be a part of the web, by means of using existing well-known standards. This way, the basics of WoT imply the use of web services programming APIs as REST (Representational State Transfer) [8], standard protocols as HTTP (Hypertext Transfer Protocol) [9], and communication technologies as WebSockets [10]. These elements are part of the application layer, which simplifies the building of applications involving the IoT [6].

A first clear approach of this paradigms is the "Distributed Computing" context. This was the context of our first generation of remote laboratories, based on the IoT market. In order to achieve better learning results, the students had to program real low-cost IoT devices, managed by Raspberry Pi and Arduino Yun platforms, which were installed inside a scale mock-up of a smart house. They had to integrate them into the cloud using the WoT model. These innovative technologies are been applied inside the learning model of many courses [11,12], allowing students a smooth and natural approach to the previous mentioned technologies and their diverse applications [13–17].

This first experience showed that there are more applications for this approach, aside from the "Distributed Computing" context. Our first generation presented some drawbacks. For instance, the creation of new experiments was ad-hoc. A computational load-balance system was also needed to avoid the fall of the services. It should take into account that the execution of processes takes place on low-cost IoT devices; these devices have reduced computational capabilities. Finally, lecturers cannot easily track the students' performance. They must search among logs to determine what actions were carried out. As an extra requirement, we were also searching for a platform that can handle interdisciplinary practical experiments from diverse contexts. Thus, it was a clear necessity to find a new platform that provides these features. Despite all these issues, WoT remote laboratories are effective educational resources, as the previous research concluded. One of its advantages was the possibility to generate a secure and closed environment for students. This fact turned our attention towards the "Cybersecurity" topic.

Among the various problems arising from the current digital society in which we live, cybersecurity has become a problem and a fundamental challenge. The exponential need for cybersecurity professionals has been growing faster than for trained qualified professionals [18]. Our future engineers must be able to address the possible technological threats of the internet, not only in a theoretical but also practical way. This approach helps the development of critical thinking skills [19–21] and gamification [22–24], among other topics of interest, as a lower-level support infrastructure.

This challenge is very ambitious, since each student's environment must take place in a controlled independent environment, which guarantees isolation properties from the rest of the infrastructure, and avoiding collateral incidents. For this reason, a natural evolution of our laboratories was to move towards a more advanced safe environmental infrastructure, paying attention to the cybersecurity topic. This is known in our work as the second generation of IoT labs. To support this evolution, a new multi-paradigm platform has been created with several learning contexts, Laboratories of Things at the Spanish National University for Distance Education (LoT@UNED) [25], which runs over a network of IoT devices, combined with cloud services to handle storage and scalability in an efficient way.

This research shows our experiences using these kinds of technologies, and it has lead us towards two generations of remote laboratories based on IoT devices, called Laboratories of Things (LoT). Both of them present support for internet connection and enough computation power to safely run different-activity contexts, in terms of layer software. Both hardware/software platforms avoid tedious preparations (by the students) of environmental setups for the activities. They are directly purposed towards the tasks of the activity.

Therefore, in the first generation of LoT, we have designed and developed a home automation based-learning system, following the WoT philosophy. Our students use the system to take on the challenge of acquiring knowledge in cloud computing solutions to control and supervise things—IoT devices—which are integrated into the internet using web technologies. This context provides enough

degrees of freedom, so that the students can implement different approaches to solve the proposed activities by the teaching team.

Students have available a collaborative learning environment, along with the distance education platform of the Spanish National University for Distance Education (UNED). This platform has been already used with real and satisfactory learning experiences [25–27]. Our second generation of laboratories presents a fully-functional platform (LoT@UNED) that runs on IoT devices. The LoT@UNED platform has expanded the domain of the experiments that can take place in it.

This paper is structured as follows. A description of our first generation of IoT laboratories for distributed computation is given in next section. Section 3 describes our second generation of IoT laboratories, and its application to cybersecurity. In Section 4, a learning effectiveness analysis is presented and the obtained results are discussed. Finally, Section 5 draws the main conclusions of this work.

## 2. First Generation of IoT Laboratories

### 2.1. The Distributed Computing Context

The distributed computing subject belongs to the Communication, Networks, and Content Management post-degree of UNED. Students must learn programming techniques focused on web service solutions and applications. These techniques are based on REST and SOAP (Simple Object Access Protocol) [28] protocols and their implementations in cloud infrastructures: PaaS (Platform as a Service) and SaaS (Software as a Service) [29–31]. It is well-known that cloud computing is a new paradigm that promotes the sharing of resources, dynamically allocated on demand by multiple end-users. This paradigm allows rent software and hardware from a cloud provider when it is needed, maximizing the economic benefit and minimizing maintenance costs [32]. Further details about this deployment and the results of its application can be found at [25].

### 2.2. Description of Things

The physical objects ("things") are distributed within the building of a scale mock-up which depicts a home with three rooms with individual lighting, video streaming, and heating management by temperature (See Figure 1).

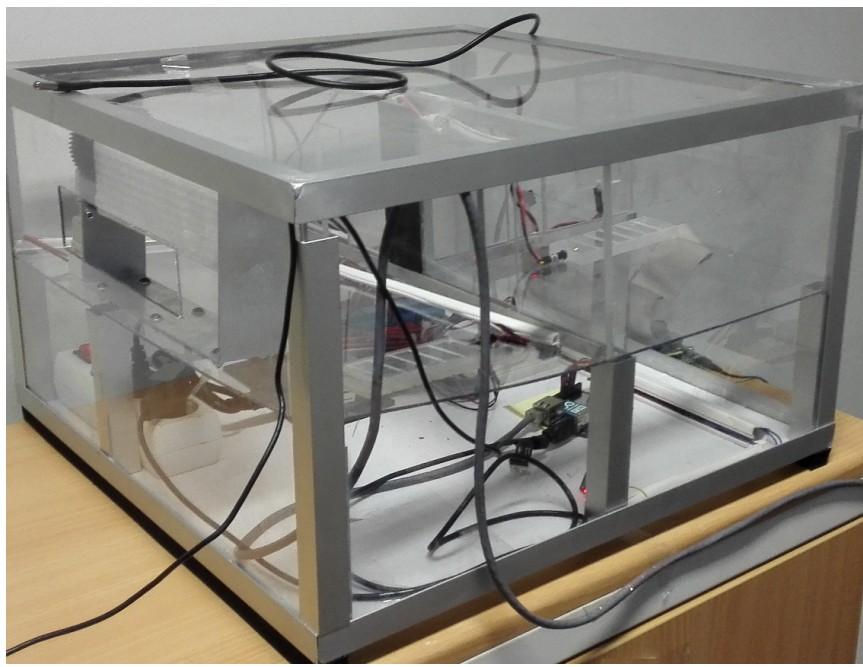

**Figure 1.** Scale mock-up of a smart house (extracted from [25]).

In order to monitor, capture, and manage the capabilities of the home things, two web platforms has been developed, in the line of [33]; one for the things connected to the Arduino Yun (located inside the house) and the another one for the things connected to the Raspberry Pi (located outside the house). So, our students have a completed vision of state of smart house things, as we can see in Figures 2 and 3, for both hardware platforms, respectively.

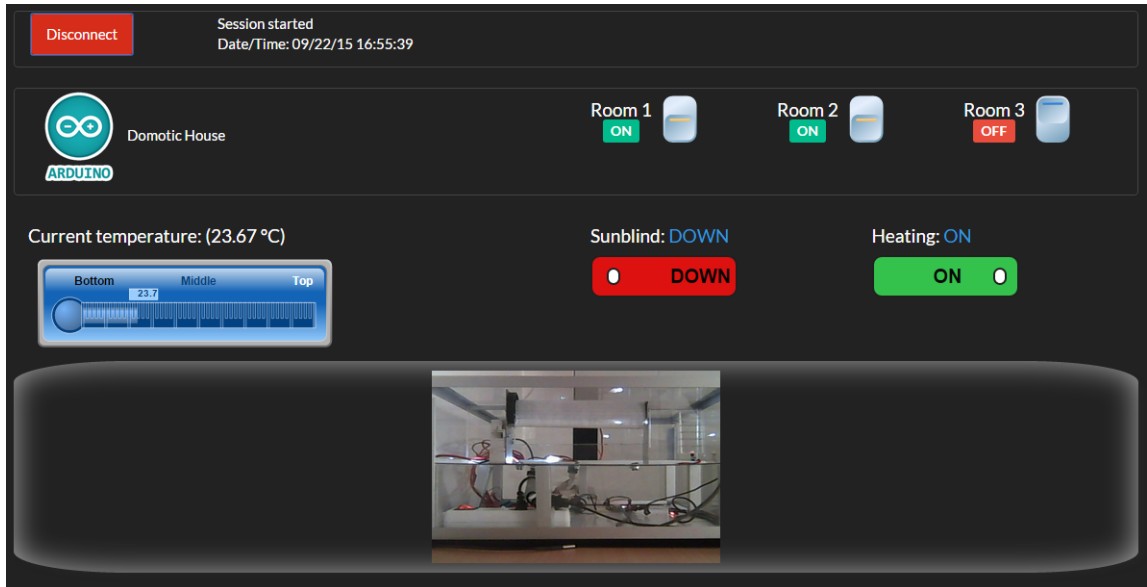

**Figure 2.** Web platform for Arduino Yun (extracted from [25]).

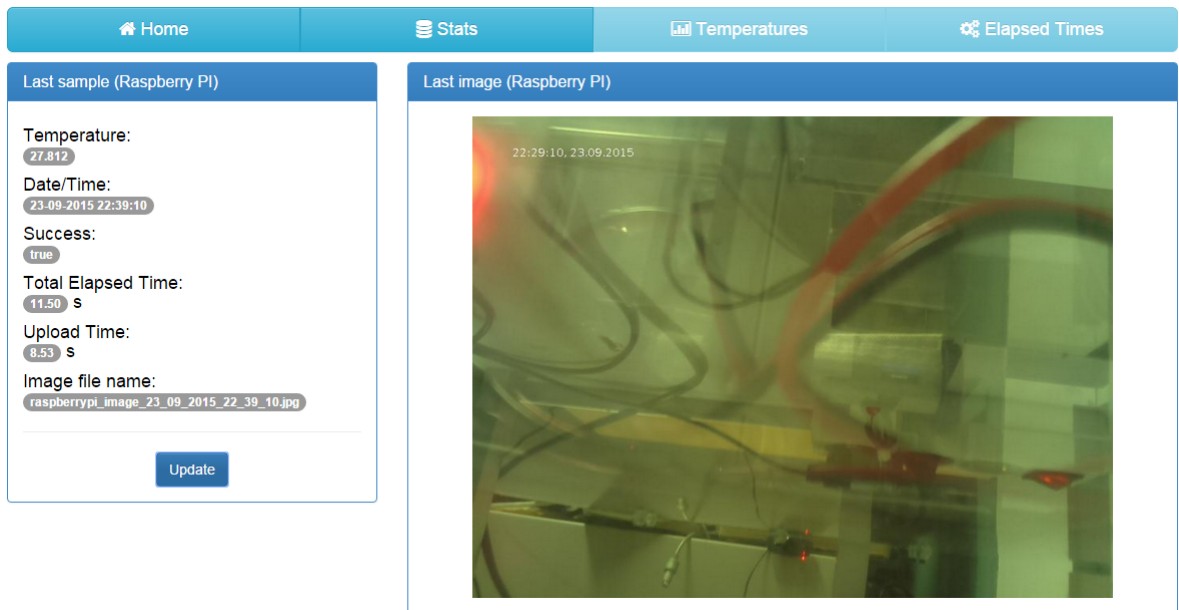

**Figure 3.** Web platform for Raspberry Pi (extracted from [25]).

Table 1 contains the list of available things in the home that can be accessed by the students. As we can see, the number of things is wide enough to develop a home automation application very close to a real one. Furthermore, the hardware platforms (Raspberry Pi and Arduino Yun) that manage these things have free hardware resources, to manage more things in future upgrades: Atmospheric pressure sensors, relative humidity sensors, gas sensors, fans, and so on. This way, the house could become more controllable and smarter: More comfort, advanced security, health assistance, energy saving, and so on.

**Table 1.** Home List of Things.

| Thing | Connected to | Function |
|---|---|---|
| LED 1 | Arduino Yun | Room 1 light |
| LED 2 | Arduino Yun | Room 2 light |
| LED 3 | Arduino Yun | Room 3 light |
| Heater | Arduino Yun | Heating for the home |
| Sunblind | Arduino Yun | Natural light (sunlight) |
| Temperature Sensor 1 | Arduino Yun | Inside temperature |
| Temperature Sensor 2 | Raspberry Pi | Outside temperature |
| Camera | Raspberry Pi | Home remote video streaming |

The reason for using these two different hardware platforms is to take advantage of characteristics supplied by both of them, and to include these very used, embedded open-source hardware systems in educational environments [34], at our student's disposal. Moreover, they have already been used in diverse designs of home automation [35–37].

From the point of view of development technologies, the WoT paradigm uses these web services to expose the things to the world by a REST API that transforms them into programmable things. Therefore, the hardware platforms that are responsible for handling at low level, the things must be able to run these web APIs. Finally, to integrate the WoT model into the cloud, several external software and services have been used by our students as the service for sharing media files (photos).

In order to fulfill the requirements for student's tasks, some additional services are needed. In the case of Raspberry Pi platform, a set of services has been developed to provide access to a "sample" of information for the things: Camera image URL (stored in a SaaS provider for media files named MediaFire [38]), "outside" temperature value, date, and elapsed times in milliseconds (total time and time involved in the uploading of image to the SaaS provider). Additionally, the historical saved data (which are stored in the Raspberry Pi) could be fetched, to get more sample information.

In the case of the Arduino Yun platform, another set of endpoints (a REST API) are available to be used by students. There is one endpoint to get a "sample" of the smart house current status; likewise the case of the Raspberry Pi platform. This "sample" includes information about the status of lights (true is on, false is off), heater (same as lights), last operation with the sun-blind ("up" or "down"), "inside" temperature, time-stamp for data, and if the house is being managed (session parameter). The endpoints allow management of the "things" in the smart house. This way, the two WoT platforms are opened to the internet, in order to be consumed by developers (in this case, the students).

## 3. Second Generation of IoT Laboratories: LoT@UNED

### 3.1. Cybersecurity Context

The cybersecurity subject belongs to Computer Science Engineering, and consists of 6 ECTS (European Credit Transfer System) credits. This subject deals with the area of information security from a practical point of view, not focusing on physical and electronic security. The methodology of teaching/learning has been adapted to the context of the European Higher Education Area (EHEA) [39]. This methodology implies much more periodical virtual attendance and interaction with/among students than for the traditional learning/teaching process. This is even more noticeable in the case of the UNED, because of the high number of students and its own distance methodology. Therefore, the main tool used by students/lecturers is the e-Learning platform aLF [39]. This online course have communication and assessments' tools and tracking information of students.

In particular, the cybersecurity subject was designed by following the next competences/qualifications (Q):

Q1    Capability to design, develop, select and evaluate applications and computer systems, ensuring their reliability, safety, and quality, in accordance with ethical principles and current legislation and regulations.

Q2    Capability to design the technical specifications of a computer installation that complies with current standards and regulations.

Q3    Capability to choose, design, implement, integrate, evaluate, exploit, and maintain hardware, software, and network technologies, within the parameters of cost and quality.

In order to acquire the previous qualifications, the subject has the following objectives defined:

OBJ1    Understand the importance of introducing (or not) cybersecurity as a design criterion in any information technology system.

OBJ2    Understand the most common current problems of lack of cybersecurity in information systems, applications, and networks.

OBJ3    Classify the different attacks from the point of view of thread, impact in the organization, and likelihood of occurrence.

OBJ4    Understand the need for the implementation of a cybersecurity policy in any organization.

OBJ5    Being able to implement the basic cybersecurity defenses in operating systems, applications, and basic communications devices.

OBJ6    Being able to apply the most basic concepts learned, related to cybersecurity in networks, systems, and data, to a specific organization.

OBJ7    Understand what firewalls and cybersecurity scanning tools are, how they are used, and what role they play in a cybersecurity policy.

OBJ8    Understand intrusion detection systems (IDS) and what role they play in a cybersecurity policy.

From the qualifications and objectives of the subject, the inclusion of practical activities is fundamental. Lecturers must provide students with controlled environments to achieve the proposed objectives and, also, do not put at risk the underlying infrastructure.

From the proposed qualifications and objectives, students have to develop three significant practical cases or activities during the period of the subject. The cases are the following ones:

A.    The first case is based on an analysis of network traffic. Students must try to detect if any type of attack is going on.

B.    The second case is oriented towards the correct configuration of a firewall in a specific context.

C.    The third case scenario is based on the configuration of an Intrusion Detection System, in order to detect a particular type of malicious activity.

Additionally, students must take a face-to-face exam in order to pass the course. This exam is made up of a questionnaire and a practical exercise, similar to those developed during the course.

The experience of making a practical assumption in a controlled context through the remote laboratories of LoT@UNED was carried out in the second practical case (B) of the subject. This is oriented to the configuration of firewalls, which is composed by the following tasks:

A1.    First, they are asked to think about the case. Thus, as a result of this reflexive phase, they must design a security policy associated with the desired security configuration of the firewall.

A2.    Afterwards, they must translate the designed policy into the real firewall rules inside the LoT@UNED platform.

A3.    Once the practical phase is finished, they have to sum up in a report the reached conclusions. This report is graded by lecturers.

To prepare the activity, a detailed guide to the context of the case study was provided to the students. Also, information about the configuration of the firewall and the instructions to get access to

the practice platform were delivered to students (all in PDF format). In addition to this, the teaching team developed several video-lessons. First video explains the theoretical concepts related to firewalls. Second video-lesson is an example of configuration of the firewall, similar to the practical case they had to develop.

### 3.2. LoT@UNED Architecture

Figure 4 shows an outline of the main components of the LoT@UNED platform. This platform is divided into two major components: a user/session manager and a virtual console that allows interaction with "things". The user manager allows users to log in with the corporate UNED account (students and teachers), as well as control access to the virtual console. Using this virtual console, students can perform different types of practice through the "virtual" execution of commands from a Linux system, knows as scripts. These commands do not really run on the provided virtual console, but directly on Raspberry Pi 3 devices, through a service orchestration platform. These devices are associated with this platform through a broker provided by the IoT service available on the IBM cloud platform (named IBM Cloud [40]). The entire system relies on the MQTT [41] protocol. To guarantee the persistence of the sessions, with the aim of analyzing and grading scripts, a Cloudant database [42] is used (a persistence service in the IBM Cloud).

The deployment of different services over a cloud provisioning system (IBM Cloud) allows the implementation of scalable mechanisms by adding different IoT devices and/or extending the existing ones (horizontal/vertical scaling). When using low-cost IoT devices, it is necessary take into account the computing load, due to its reduced computational capabilities. Thus, a balanced growth and management of the different components should be implemented. Additionally, better availability of the same services in the different IoT devices allows to increase the use of the system in terms of concurrent sessions.

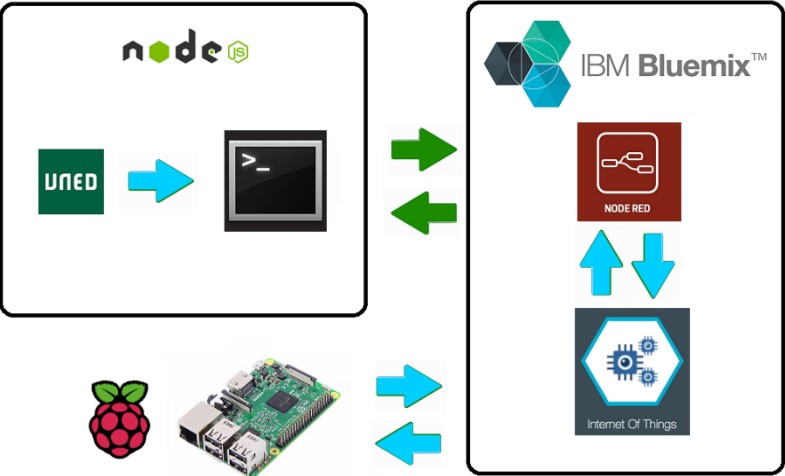

**Figure 4.** Architecture of the LoT@UNED platform.

The main features of this platform include:

1. The login can be done directly with the user credentials of UNED, without the need to be registered in advance.
2. It provides an interactive console, as can be seen in Figure 5, based on commands from the Ubuntu distribution, which allows its direct execution on devices.
3. Robustness related to network problems (self-recovery). Therefore, if the user session is unintentionally disconnected, its progress is recorded in the system. The session can be resumed at the same point in which the student left after re-establishing the connection.

4.　Depending on the context determined by the practice, the system allows the user only to execute a certain set of commands. The platform has contextual help that reports the student the available commands, as well as specific links to the available documentation of these commands.

5.　The temporary extension of the sessions of the students can be limited (challenges directed by temporary consumption). The student is informed by a stopwatch in the console.

6.　At the end of a session, students obtain a report with the tracking information of their sessions. This report can be downloaded in PDF format. In the same way, the professors associated with the practice can examine the corresponding reports to all the students who have carried out the practice in the platform.

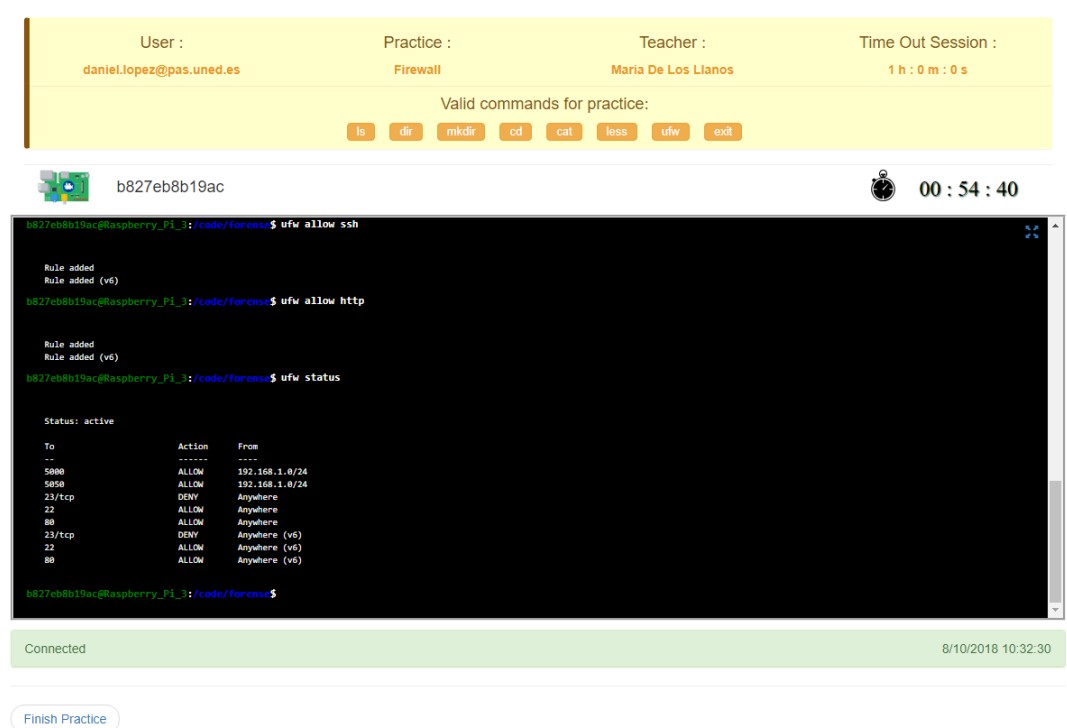

**Figure 5.** LoT@UNED Virtual Console appearance.

### 3.3. Comparison among Generations

Prior to the use of the LoT@UNED platform, the environment for the deployment of the cybersecurity activities was more manual. Lecturers used two interconnected Raspberry Pi, seen as low-cost computers. One of them had the role of a SSH server for the students' access. The another one was the target server, in which students should harden a suitable firewall configuration. Students had to request lecturers, by email, their corresponding SSH credentials. Lecturers had to create the associated credentials. This approach also lacked support for concurrent sessions. Only one of the students was able to work with the infrastructure at the same time. Thus, students had to ask for laboratory free-slots with the help of a calendar. Once a student finished his/her slot of time, lecturers had to reset all devices. To sum up, this first approach was more tedious for both students and lecturers.

There were several improvements thanks to the LoT@UNED platform. As a starting point, lecturers only have to create a Docker container as an experiment setup. LoT@UNED automatically registers the container, and several different activities can be done within it. Table 2 summarizes the main differences among these two generations.

**Table 2.** Comparison among different properties of the presented two generations.

| Feature | 1st Generation | 2nd Generation |
|---|---|---|
| Experiment creation | Ad-hoc procedure | Automated procedure supported by the platform |
| Experiment access | SSH user provided by lecturers | Easy access with the Institution credentials |
| Management facilities | Manual | Automated |
| Tracking students' activity | Direct log analysis | PDF report provided by the platform |
| Number of supported experiments | One | Several |
| Concurrence | One student at time | Up five students at time |
| Disciplinary | One course | Multidisciplinary |

## 4. Learning Effectiveness Analysis

The principal purpose of this research study is to explore the impact of using a set of IoT devices as a technological environment, in order to analyze the effectiveness of the learning process in the context of cybersecurity. To this end, and based on the state of the art, the LoT@UNED platform [25] was designed and deployed in the University infrastructures.

Our experiments were conducted to address the following hypotheses:

**Hypothesis 1 (H1).** *The use of the LoT@UNED platform improves the learning engagement of students, and their learning outcomes.*

**Hypothesis 2 (H2).** *Students' effort by using the LoT@UNED platform is higher than with a classic approach.*

**Hypothesis 3 (H3).** *Students' academic performance is better with the LoT@UNED platform in contrast to a classic approach.*

To achieve our objective, the students' experience is presented and exhaustively analyzed and discussed, with and without the use of the LoT@UNED platform, during their learning process.

### 4.1. Participants and Materials

The current study was conducted with the second generation of IoT laboratories (that is, the LoT@UNED platform) in the cybersecurity undergraduate level subject, taught at UNED in the second semester of the 2017–2018 academic year. The amount of students enrolled in the subject was 238. Among respondents (the 54.2% of students), 115 out of 129 were male and 14 out of 129 were female, as observed in Table 3, for the 2017–2018 academic year. Some of the results will be analyzed and discussed compared with the data gathered in the previous 2016–2017 academic year, when a traditional approach was employed (with a distance methodology). For this reason, Table 3 also shows the corresponding demographic data for the 2016–2017 academic year.

From the academic performance point of view in the 2017–2018 academic year, only 57% of students successfully passed the course, and 72% of students had performed all the evaluation activities throughout the course. Whereas, during 2016–2017, 246 students were enrolled in the subject. In total, 64.6% of the students successfully passed the course, and 68% of students had performed all the evaluation activities during the 2016–2017 academic year. The students' academic goals, in terms of competence and evaluations, are given in the cybersecurity context from a previous section.

As previously described, the purpose of this experience has been the implementation of a firewall within the system, carrying out the execution of a series of configuration rules. To achieve this, students had to log-in, book a session, and solve this practical activity. The maximum time-slot for each session was one hour. Students could take as many sessions as they considered necessary to solve the activity. The period to complete the activity was 2 months, but the deadline was extended by two more weeks.

**Table 3.** Demographics of the questionnaire respondents.

| Demographic Course | | 2016–2017—Amount (%) | 2017–2018—Amount (%) |
|---|---|---|---|
| Occupation | Non-computer science related job position | 100 (54%) | 65 (50.38%) |
| | Computer science related job position | 43 (23%) | 25 (19.39%) |
| | Other situation | 42 (23%) | 39 (30.23%) |
| Gender | Male | 167 (90.3 %) | 115 (89.15%) |
| | Female | 18 (9.7%) | 14 (10.85%) |
| Age group | ≤30 years | 54 (29.19%) | 39 (30.23%) |
| | 30–40 years | 95 (51.35%) | 55 (42.64%) |
| | 40–50 years | 31 (16.76%) | 31 (24.04%) |
| | ≥50 years | 5 (2.7%) | 4 (3.10%) |
| Familiarity with Cybersecurity | Very unfamiliar | 46(24.86%) | 37 (28.68%) |
| | Unfamiliar | 50 (27.03%) | 24 (18.60%) |
| | Neutral | 40 (21.62%) | 31 (24.03% ) |
| | Familiar | 32 (17.3%) | 16 (12.41%) |
| | Very Familiar | 17 (9.19%) | 21 (16.28%) |
| Voluntary use | Strongly Agree | 95 (51.35%) | 68 (52.71%) |
| | Agree | 53 (28.65%) | 24 (18.61%) |
| | Neutral | 15 (8.11%) | 22 (17.05%) |
| | Disagree | 12 (6.49%) | 13 (10.08%) |
| | Strongly disagree | 10 (5.41%) | 2 (1.55%) |

*4.2. Instruments and Data Collection Procedures*

When a student enrolled in the subject, he/she was asked to fill in a questionnaire with occupation, demographic information, and other factors related to the participants in the experience, in order to get student profiles. This questionnaire contained information about students' sex and age, and they were asked to specify their job occupation, familiarity with cybersecurity, and if they are open to the use the LoT@UNED platform. More details are given in Table 3.

After that, students accessed the theoretical and practical resources, hosted in the learning platform. Each practical activity started and ended in a particular period of time within the course. Once a practical activity started, students were provided with a guide, a PDF document, and a dedicated YouTube video-lesson, linked to the virtual space. The video-lessons on the YouTube platform provided us with significant statistics about the duration of the students' views, and so on. The practical activity score was scored by lecturers, after assessing the students' reports and the trace of the sessions recorded at the LoT@UNED platform.

On the other hand, students were encouraged to answer an opinion survey after the period of the practical activity. The format of this questionnaire was based on UTAH methodology [43–45] for evaluating the user acceptance of a new technology.

After the period of the subject, students performed a face-to-face exam in order to pass the subject. Both activities and exam scores were compared, to determine if there was an improvement in the global learning process.

Therefore, the interactions of students with the LoT@UNED platform were recorded and stored for statistical analysis. These include learning and device data:

1. Downloaded and accessed content. All interactions of students with video lessons and documents were recorded for the study.
2. Opinion survey to measure the users' acceptance of technology.
3. Student marks/scores, from both practical activities and the face-to-face exam.
4. Device data, composed of log files from IoT devices, to calculate an effort estimation of students.

The LoT@UNED platform records each command issued by a student with the corresponding timestamp. Thus, lecturers can estimate the effective time spent in each session. This report is available for both lecturer and student.

**Table 4.** Comparison between both course editions.

| | 2016–2017 | 2017–2018 |
|---|---|---|
| Course materials | Course documents (PDF), video-lesson (YouTube), and recommended book | Course documents (PDF), video-lesson (YouTube), and recommended book |
| Activity materials | Activity guide (PDF) and video-lesson (YouTube) | Activity guide (PDF) and video-lesson (YouTube) |
| Course structure | • 12 course lessons.<br>• Three practical experiments: network trace analysis, firewall configuration, and IDS configuration.<br>• Additionally, students are provided with solved experiments and self-assessment questionnaires. | • 12 course lessons.<br>• Three practical experiments: network trace analysis, firewall configuration, and IDS configuration.<br>• Additionally, students are provided with solved experiments and self-assessment questionnaires. |
| Activity structure | 1. Student reads activity materials.<br>2. Student requests lecturer SSH credentials.<br>3. Student connects to the remote laboratory and performs the activity.<br>4. Student reports lecturer he/she has finished the experiment.<br>5. Student makes some conclusions about the experiment and write a report about it.<br>6. Student answers an online quiz. | 1. Student reads activity materials.<br>2. Student designs the access policy for the exercise.<br>3. Student accesses to the remote laboratory with the subject credentials.<br>4. Student implements the designed access policy.<br>5. Student makes some conclusions about the experiment and write a report about it. |
| Students' outcomes | Activity report and final questionnaire | Activity report |
| Communications' tool | Forum at learning platform | Forum at learning platform |

The comparison between both course editions is given in Table 4. Several indicators (materials, structure, student outcomes, and communication tools) are summarized for each academic year, with and without the employment of the LoT@UNED platform.

### 4.3. Data Analysis

This analysis approach is based on a mixed search approach, known as the sequential explanatory design method [46,47]. A sequential explanatory design method is considered a legitimate, stand-alone research design in engineering education, since it combines the strengths of both qualitative and quantitative features [48,49]. This research method is composed of two phases: A quantitative phase, followed by a qualitative phase [50]. For the quantitative one, data about the interactions of students by using the LoT@UNED platform, surveys, and so on, were collected, and then statistically analyzed using parametric and non-parametric techniques. For each quantitative indicator, a corresponding qualitative criteria is defined, in terms of the hypotheses.

In order to validate the considered hypotheses for this research study, Table 5 depicts a set of evaluation criteria based on the selected indicators in a qualitative way, the corresponding quantitative indicators, and the source of the information to gather these qualitative and quantitative indicators.

First, it is possible to know whether the use of the LoT@UNED platform improves the learning results and student engagement (defined by H1), by means of the results obtained from the evaluation questionnaire in the second semester of 2016–2017 academic year, in comparison with the ones in the second semester of the 2017–2018 academic year. Furthermore, in relation with student engagement, we expect to have an indicator that shows this engagement after the inclusion of real IoT devices in the course.

Since our students are immersed in a distance education environment, there is an approximate way to know whether the IoT technology gains student attention by analyzing his/her access to the

course materials and the analysis performed on the final survey. As the students are provided with online videos hosted on YouTube, this platform offers us a detailed range of statistics related to the student retention and visualizations of the video.

In order to get student's effort for using a new technology in the subject, it is necessary to know the period of time that students need to complete an activity session properly. In particular, the spend time/total time rate gives us a quantitative approach of this effort. Using this information, it is possible to make a comparison between data of the activity hosted in the LoT@UNED platform and the previous case. Thus, we know whether the student has needed a higher effort, proposed in the H2 hypothesis.

Finally, it is also interesting to know whether the academic performance improves the global course level. Therefore, we can draw the impact of use of WoT in courses with specific educational competences about cybersecurity. The classic model based on predictable data (static vectors or arrays of properties) is changed by others based on non-predictable data, which come from the IoT things in real-time. This way, it is possible to study and analyze the student's response in terms of global scores, which are used as indicators to evaluate the H3 hypothesis. In this case, we compare the results obtained between the 2016–2017 and 2017–2018 academic years for the same activity.

**Table 5.** Summary of the employed evaluation criteria, quantitative indicators, and data gathering.

| Qualitative Evaluation Criteria | Quantitative Indicator | Data Gathering Source |
| --- | --- | --- |
| Students' performance in tasks which the use of WoT platforms is involved (H1) | Task questionnaire evaluation | e-Learning Platform |
| Students' interest in social activities related to WoT and IoT devices (H1) | Material accessed | Youtube Analytics |
| Students' effort in using the WoT platform during related tasks (H2) | | LoT@UNED |
| Students' acceptance of the new platform | Opinion survey | e-Learning Platform |
| Students' academic performance (H3) | Final score | e-Learning Platform |

For every quantitative indicator, a test of normality distribution will be performed. When data is correct, in terms of normality distribution, a *t*-test (paired or 2-sample) will be executed. All tests were conducted using an alpha level of 0.05.

*4.4. Results and Discussion*

In order to investigate whether the LoT@UNED platform had a positive impact on the students' learning engagement and outcomes, the students' effort by using the platform, and their academic performance when compared with a traditional approach, the gathered results are analyzed and discussed, both from a quantitative and qualitative point of view, according to the previously defined hypotheses.

**Hypothesis 1 (H1).** *The use of the LoT@UNED platform improves the learning engagement of students and their learning outcomes.*

For the analysis of the students' learning outcomes, a Shapiro-Wilk test [51] with a normal distribution was performed to examine the distribution of the scores achieved by students, for both of the 2016–2017 and 2017–2018 academic years, including the mean (M) and standard deviation (SD) values. The obtained scores were (M = 7, SD = 1.86) and (M = 8.4, SD = 0.71), respectively. More than one point, by mean. See Figure 6, where the histogram graphs represent data distribution for both academic years. The vertical axis represents the number of students inside the bin, and the horizontal axis is the grade bins.

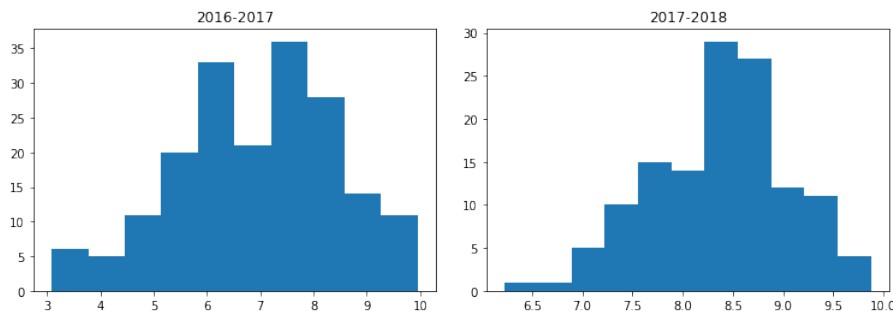

**Figure 6.** Histograms for the score distribution of the activity.

The results of these tests indicate that data may come from a normally distributed population (W = 0.927 *p*-value = 0.05 for the 2016–2017 academic year; W = 0.729 *p*-value = 0.176 for the 2017–2018 academic year). For this reason, a paired *t*-test [52] was used, to compare the scores before and after the introduction of the LoT@UNED platform in the subject. The results of this test ($t = -7.46$, *p*-value< 0.003) indicated that there is a statistically significant difference between the student scores of both academic years. For this reason, it can be concluded that students improved their comprehension of how a firewall is configured.

In order to get a measure of the learning engagement, the "virtual" participation in related resources accessed by students was analyzed. Again, the 2016–2017 and 2017–2018 academic years are compared, in order to test the benefits of using (or not using) the LoT@UNED platform. As described before, the supporting video-lessons were hosted on the YouTube platform, since it offers us detailed statistics about the visualization of the video (see Table 6, for further details). For instance, the mean period of time in which a video was visualized is higher for the 2017–2018 academic year, 39.9% (8:18 out of 14:37 min) instead of 32% (5:13 out of 20:50 min), due to student interest. Visualization, visitors, and interaction also increased. It is worth remarking that that only 2% of the visualizations took place outside of the aLF platform.

**Table 6.** Video-lesson main statistics of the students' visualizations.

| Academic Year | Total Duration | Visualizations | Average Visualization Time per Session in minutes (and % of Duration) | Unique Visitors | Interactions |
|---|---|---|---|---|---|
| 2016–2017 | 20:50 min | 253 | 5:13 min (32%) | 109 | 9 |
| 2017–2018 | 14:37 min | 260 | 8:18 min (39.9%) | 120 | 11 |

A Shapiro-Wilk test with a normal distribution was also performed of the video-lessons visualizations. The test results indicated a normal distribution of data in the multimedia resources (W = 0.785, *p*-value = 0.01 for the 2016–2017 academic year; W = 0.40, *p*-value = 0.028 for the 2017–2018 academic year). Therefore, a *t*-test can be used to determine whether there was a statistically significant difference between the visualization of the video-lessons. The *t*-test results ($t = -1.027$, *p*-value = 0.003 for multimedia resources) can be used to conclude that there was a significant difference among them. As observed in Figure 7, there was clear a increment of the average minutes that students spent watching video-lessons, from 5:13 min (32% of the video duration) during 2016–2017 to 8:18 min (39.9% of the video duration) during 2017–2018. Thus, student visualizations of the video-lessons were higher, with more visual retention, when the LoT@UNED platform was used for practical activities. As a conclusion, there was higher interest in the multimedia resources.

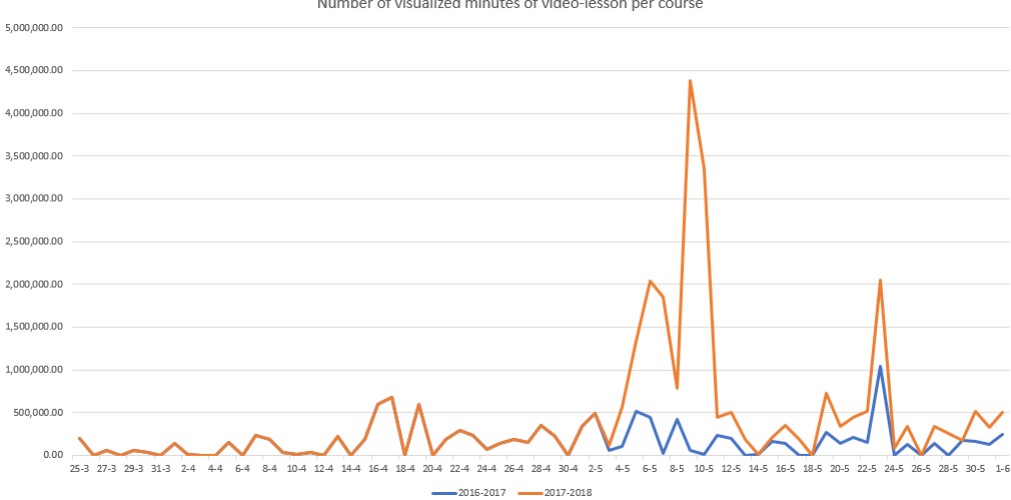

**Figure 7.** Number of minutes visualized (per day) of the associated video-lesson during the 2016–2017 and 2017–2018 academic years.

From the detailed quantitative results, it can be concluded the H1 hypothesis was proved. The analysis of the students' learning outcomes was also done through the qualitative features from an opinion survey about their satisfaction with LoT@UNED for the configuration of a firewall, in terms of perceived usefulness, effort needed to learn (estimated effort), attitude, social influence, easy of access, and intention to use. This discussion will be tackled later.

**Hypothesis 2 (H2).** *Student effort, by using the LoT@UNED platform, is higher than with a classic approach.*

The overall amount of time (counted in hours) spent, in the case of the firewall configuration, is a set of collected data from students. This data is available for both of the 2016–2017 and 2017–2018 academic years. Therefore, we can utilize this data to get the mean values, and make an analysis to determine their significance.

During the 2016–2017 academic year, according to the information hosted in the virtual platform of UNED (aLF), each average session lasted almost an hour (M = 59 min, SD = 14), with only one session per student. This was recorded in the access logs (by SSH) of the remote virtual machine used to carry out the A2 task of the practical case. On the other hand, during the 2017–2018 academic year, the LoT@UNED platform had registered 520 sessions, with an average of 8 daily-sessions and 4 sessions per student. As observed in Figure 8, the volume of sessions was concentrated in the final period of time for the subject, in some cases even reaching more than 40 daily-sessions. From these exploratory values, we can conclude that that students had spent more time doing the practical activity than in the 2017–2018 academic year.

It deserves to be highlighted that the recorded times are only related to the previously described A2 task. The A1 and A3 tasks were not measured, although we estimate that the times which are used in our computations are only a one-third part of the whole spent time. This estimation allows us to plan the activity and the posterior phase of conclusions.

First, a Shapiro-Wilk test of normal distribution was performed to examine the distribution of the estimated times (in hours) involved by student for the A2 task during 2016–2017 and 2017–2018, respectively. The test results showed that the data may have come from a normally distributed population: W = 0.952, *p*-value = 0.261 for 2016–2017; and W = 0.946, *p*-value=0.590 for 2017–2018. For this reason, a *t*-test was used to compare times in both periods. These results (*p*-value < 0.003, $t = -2.65$) indicated that there was a statistically significant difference between the minutes involved in the development of the A2 task per student in 2016–2017 academic year (M = 59, SD = 14) and 2017–2018 academic year (M = 240, SD = 6.24).

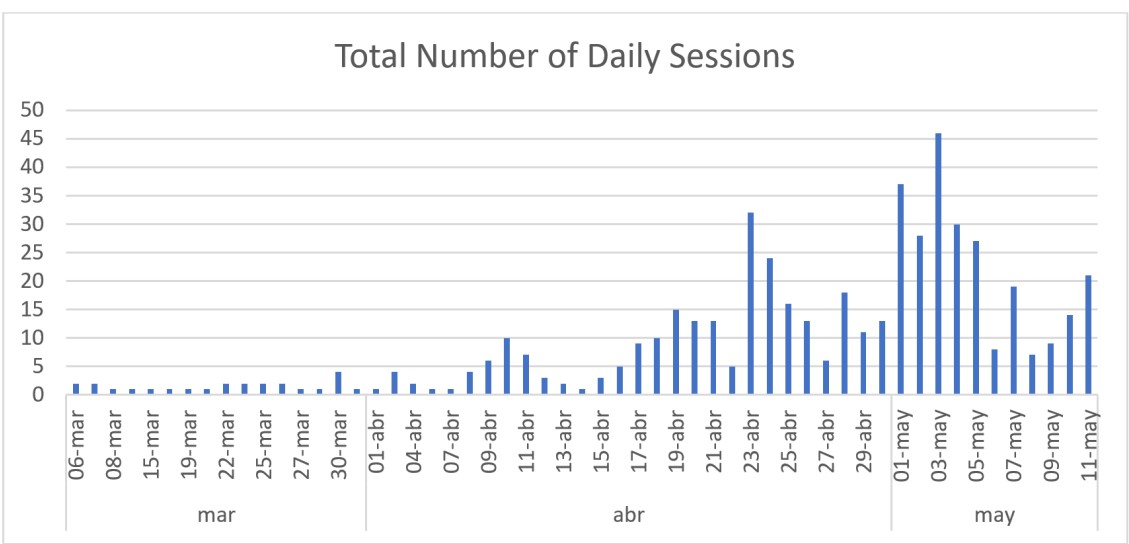

**Figure 8.** Evolution of the practical sessions throughout during the period of the subject for the 2017–2018 academic year.

The H2 hypothesis states that student effort was bigger when IoT things were employed. This hypothesis was formulated by taking into account that the use of unknown technologies lead to more time spent by the students to assimilate them, as detailed in Figure 8. According to the recorded logs of LoT@UNED, students spent more sessions preparing the activity, due to the open nature of its definition. Thus, they explored several ways of solving it. Whereas, in the previous course, they had to focus their attention on preparing the activity context (a network of virtual machines to run the firewall), and they had not explored multiple solutions. Using the proposed Lot@UNED platform, students had more available time to practice directly with the firewall and its configuration options, instead of preparing the activity context. Although, during the 2017–2018 academic year, students spent more time doing the activity, their main effort was concentrated in the activity objectives.

Additionally, students were asked their opinion about the LoT@UNED platform. The survey was a questionnaire, including four choice questions. Each statement is a five-point liker-type scale, ranging from (1) "strongly disagree" to (5) "strongly agree". A summary of the results of this survey is represented in Table 7. These questions were focused on the following indicators:

- Perceived usefulness by the students using the platform and their experience.
- Perceived effort needed to learn how to use the platform according to students' opinion; that is, the ease of use of the tools involved in the taken experience.
- Attitude towards the technological solution used, assessing whether the students perceive that using the system is beneficial or not for the purposed objectives.
- Social influence, trying to reflect how the students' opinion is perceived by other classmates and teachers about the experience.
- Perceived ease of access and perceived availability of educational resources during the experience.
- Intention of access; in other words, the students' perspective about similar experiences supported by the same platform for other experiences.

According to the obtained results from the opinion survey (see Figure 9), it is not perceived by students that the platform increases the effort of performing the activity. It is perceived by almost the 50% of students that the platform is easy of use, and it helps to improve the performance of the activity. It is not perceived that there is a social factor that influences students' opinion about the LoT@UNED platform and the presented experience. Thus, it had been perceived by almost 80% of the students that the experience was useful or strongly useful for their learning. More than the 47% of the students were willing to use the LoT@UNED platform in other activities—even in other subjects.

**Table 7.** Opinion survey results.

| Indicator | Strongly Agree | Agree | Neutral | Disagree | Strongly Disagree |
|---|---|---|---|---|---|
| Perceived usefulness | 33.33% | 44.96% | 15.50% | 5.43% | 0.78% |
| Effort needed to learn | 47.29% | 28.68% | 16.28% | 6.98% | 0.78% |
| Attitude | 47.29% | 31.01% | 16.28% | 4.65% | 0.78% |
| Social influence | 17.05% | 35.66% | 42.64% | 4.65% | 0.00% |
| Ease of access | 5.43% | 48.84% | 38.76% | 6.98% | 0.00% |
| Intention of use | 47.29% | 26.36% | 14.73% | 10.08% | 1.55% |

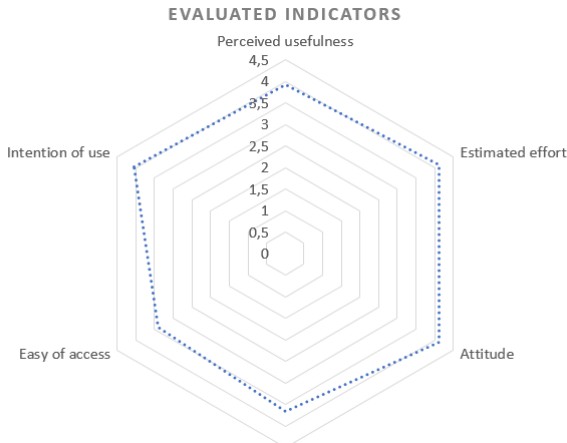

**Figure 9.** Indicators for opinion survey.

However, according to these results, the perceived ease of access and perceived availability of resources was not so satisfactory, from their point of view. This indicator can be affected due to several access problems suffered during the activity. As the activity deadline was approaching, more students were trying to access to the platform. The nearer the day to the end of the activity, the higher load of the platform was (than the rest of the course), as depicted in Figure 8. In concrete periods of time, there was not a free Raspberry Pi to perform the activity. As a consequence, some students had to wait, in order to access to the platform. Therefore, students perceive this fact as a lack of resource management availability.

As a conclusions, the H2 hypothesis is not valid, according to the students' opinion gathered from the survey, although initially it could be interpreted as proven, according to our initial statistical analysis of log-data. A deep comprehension about the time that students have spent within the LoT@UNED platform has helped us to understand that students were checking different options to solve the practical case. But, they did not perceive it as a greater effort.

**Hypothesis 3 (H3).** *Student academic performance is better with the LoT@UNED platform, in contrast to a classic approach.*

The analysis of student academic performance was done by comparison of the scores achieved by the two selected groups of students. As 70% of the qualification corresponds to the final face-to-face exam, a specific question dedicated to the three practical cases/activities was included in this. Thus, it is useful to detect cheating behaviors and, also, to increase the final score of students who had successfully performed the practical activities during the period of the subject.

A Shapiro-Wilk test of normal distribution was performed to examine the distribution of the scores for both of the 2016–2017 and 2017–2018 academic years. The test results showed that the data may come from a normally distributed population: W = 0.9, *p*-value = 0.059 for 2016–2017; and W = 0.90, *p*-value = 0.015 for 2017–2018. Taking this assumption as valid, a *t*-test was conducted to compare scores for the two academic years. The results (*p*-value = 0.004, *t* = 0.75) indicated that there is a statistically significant difference between the scores in 2016–2017 (M = 6.02, SD = 2.63) and 2017–2018

(M = 7.17, SD = 2.60). See Figure 10, where the histogram graphs represent data distribution for both academic years. The vertical axis represents the number of students inside the bin, and the horizontal axis is the grade bins.

Thus, the H3 hypothesis was proved, and so it can concluded that students' academic performance was slightly better when they used the LoT@UNED platform, in contrast to the "classic" approach.

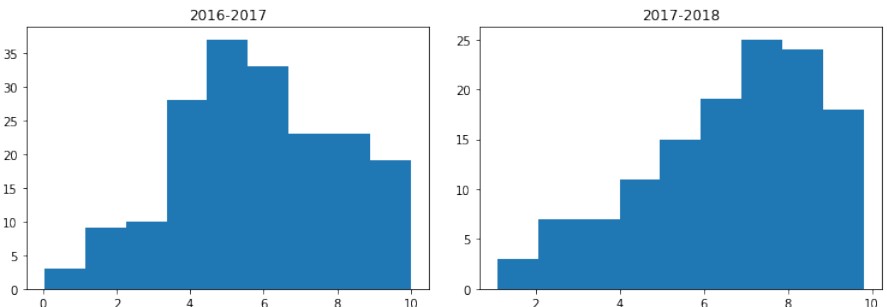

**Figure 10.** Histograms for the score distribution for the course grades.

## 5. Conclusions and Further Work

This work has presented two generations of Laboratory of Things (LoT), which represents great innovative technological achievements from the point of view of on-line education, and its natural evolution in this context. These LoT platforms improved the learning outcomes of our students, and promoted their motivation within the Computer Science disciplines. A clear indicator of this was the increase of student interactions with the resources and communication tools in the virtual courses. This statement was enforced, with the research study presented in this work for our second generation of the LoT platform—namely, LoT@UNED.

Our solution is not only focused on a fixed learning context; it is generalized and easily adaptable to support any kind of environment in the context of education, such as distributed computing and cybersecurity. This fact makes it possible to improve student learning outcomes, when compared with a traditional approach of distance methodology.

The creation of these types of activities is time-consuming from the point of view of lecturers, but the benefits for students is positively increased. The student's effort is focused on the learning objectives, instead of the preparation of complex local contexts (with virtual machines, specific containers, etc.). Thus, they can explore alternative solutions for the proposed activities in a comfortable "makerplace". This working space is composed of enough practical components to allow a better acquisition of the competences/qualifications planned for a subject.

**Author Contributions:** Data curation, L.T. and A.R.-G.; Formal analysis, A.R.-G.; Funding acquisition, R.P.; Investigation, L.T. and R.P.; Methodology, L.T. and A.R.-G.; Project administration, R.P.; Software, R.P. and D.L.; Supervision, R.H.; Validation, R.P.; Visualization, A.R.-G.; Writing—original draft, L.T. and D.L.; Writing—review & editing, Ll.T., A.R.-G., R.P., R.H. and J.C.

**Funding:** The authors would like to acknowledge the support of a research project for the period 2017–2018 from the Computer Science Engineering School (ETSI Informática) in UNED; the eNMoLabs research project for the period 2019–2020 from UNED; and the Region of Madrid for the support of E-Madrid Network of Excellence (S2013-ICE2715). Authors also acknowledge the support of SNOLA, officially recognized Thematic Network of Excellence (TIN2015-71669-REDT) by the Spanish Ministry of Economy and Competitiveness.

**Conflicts of Interest:** The authors declare no conflict of interest.

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
