# Peer review of "Web of Things Platforms for Distance Learning Scenarios in Computer Science Disciplines: A Practical Approach"

_technologies, doi:10.3390/technologies7010017_

Round 1
Reviewer 1 Report
The authors describe how a web of things platform was used in distance learning. It describes the utilization of two generations of platforms, the first for distributed computing and the second for cybersecurity. The goal is to compare how the use of the platform in the second case improved learning outcomes for the students.
The topic of the paper is highly relevant, i.e. how such a platform can be used to improve education in distance learning. Unfortunately, the current version fails to give the reader enough information for a real assessment. The introduction already indicates some lessons learnt from the first platform which is then described, followed by the second platform. However, there is no real comparison between the two, i.e. only some aspects are presented in the introduction before giving an in-depth description of each, but not followed by a more in-depth comparison. As it turns out, the main objective is to compare the performance of students in two courses, the first without using the platform and the second using the platform. While the use of the platform is clearly described, it remains unclear how it was done in the case of the previous course. This makes it practically impossible to understand the comparison. Numbers are presented that claim to show a significant difference, but there is not enough explanation how things really improve and whether they are even comparable. (What are the parameters that changed and how did they change?)
So, there needs to be a clearer description:
- What are the different tasks
- How exactly were they addressed for the first course
- How exactly were they addressed for the second course
- More qualitative explanation of the results – not just numbers
In table 4, some durations are given, but it is unclear how they are related, e.g 20:50 minutes – how are they related to the 5:13 minutes and what do you mean with 32% in this context?
In table 5 level of agreements are presented – this can be done for clear statements, but it is not clear what that means with respect to questions asked, i.e. what does “strong agreement” mean with respect to “perceived usefulness by students of the platform and experience”, as opposed to (e.g.) a clear statement: “The platform was very useful for understanding …” to which you may agree or disagree.
The paper is generally readable, but significant improvement of language aspects is needed.
Author Response
Thanks for your comments. See the attached document for the response to these comments

Reviewer 2 Report
The paper is presenting a solution for e-learning in IoT solutions. It doesn't give any relevant scientific result or architecture. In my opinion the paper should be presented in an journal or conference oriented to explain teaching solutions and its results, as this is what is included in the paper. For instance sections H1 and H2 are presenting results of how the students have been dealing and working with the platform. And also shows interfaces for teaching sessions. This is not presenting any advance over SoA or scientifc stuff.
Based on that I consider it is not appropiate for this journal.
Some comments on the paper:
L36 - laboraties --> laboratories
When you refer to WoT is not only putting data in the cloud and accesible but you also have to establish links between the things and objects. Make everything referenceable.
The introduction is going too direct into personal achievements of the authors which I personally don't like. They have to put more in context the information and what is the scope of the paper (for instance, the link with education, and what they want to achieve, similar activities, etc.). There is no SoA analysis of solutions similars to the one proposed.
The introduction suddenly starts talking about cybersecurity and changes to gamification. I think there should be a proper and smooth way of introducing the article.
L67 - There should be a reference to LoT@UNED and expand the acronym here in stead of L70.
Section 2
Nothing new is really presented. It is just a small summary that includes some sensors controlled by an arduino and a RPI. The basics to deploy a sensing system is what is explained, without giving advanced over the SoA.
There is no relation to a real web of things deployment where the systems are related and linked using for instance semantics, etc.
Section 3
It is presenting what are the items and objectives for a special lecture/course. Nothing scientific.
The architecture is well presented, but again no novelty is identified.
L261 repeated explore word.
Author Response

(The authors gave the same response as above.)

Reviewer 3 Report
The study in the paper is very relevant for the special issue on "Technology Advances on IoT Teaching and Learning". It might be of interest to understand a little more about prevention of cheating, but that is an optional suggestion. The English in the paper is, however, poor and it needs to be extensively revised by a native English speaker.
Author Response

(The authors gave the same response as above.)

Reviewer 4 Report
The manuscript is well organized and the topic is very interesting and useful. I suggest manuscript for the publication.
Author Response

(The authors gave the same response as above.)

Round 2
Reviewer 1 Report
The quality of the paper has been significantly improved and review comments have been taken into account. However, I still see the need for further improvement. With section 3.3 and Table 2, there is now a nice comparison between the two generations of platform. I’m still missing such a clear comparison between the two approaches in the second part. What kind of course materials (yes, there was video, but not much explanation), how was the course structured, what was expected from the students, was there interaction with tutors. (Yes, some information is somewhere in the text, but a table summary would be useful). In particular, there is still very little information on the traditional approach.
In Section 4.1 detailed information for the 2017-2018 academic year is given – why is the same not provided for 2016-2017?
There is still mainly the proof that we have normal distributions and a significant difference regarding the performance? Why don’t you show a graphical representation of how the actual marks have been distributed (of course still show the significance)? Is there a way to better explain the significance of the result to the average reader? What does t=-7.46 actually mean?
The video viewing percentages are still unclear to me: If the relation is 5:13 minutes to 20:50 minutes, I calculate a percentage of ~25%, not 39.9%???
I still have an issue with the way the opinion survey (Table 6) is formulated. If the question is “Estimated effort needed to learn …” – I would expect an answer like “huge effort”, “medium effort”, “little effort”. If you want the answers on an agreement scale, you would have to make a statement: “The effort needed to learn is very low” and then the agreement could be from “strongly agree” to “strongly disagree”.
There has been some improvement regarding the language, but more improvement is needed.
Examples:
- p.1: Nowadays, THE Internet is a big and complex (example of a?) globalLY scalable network” (not “sample”)
- p.1: These elements interact among themselves across ….
- p.1 ... paradigms SUCH as Pervasive Computing …
…
- p.9 ... One computer, WHICH was running an SSH server, haD the access role …
- p.9 Lecturers HD TO create …
- p.9 Thus, students HAD TO request their …
- p.9 As it can easily BE SEEN, this approach
Table 5 is not shown completely as it does not fit onto the page
Author Response
Please, see the attached PDF document with the replies to your comments

Reviewer 2 Report
When going through the new version I have more carefully gone through some of the references and I have detected that paper introduction, section 2.2 and section 2.3 are almost taken identically from reference 22 (Teaching cloud computing using ... at EDUCON 2018). This means that almost 6 pages of the current paper are taken directly from the other one. In this one, they seem to focus more on CyberSecurity teaching, which is really emphasize in testbed description.
Besides, and although the study is complementary the sections concerning data analysis in both papers (this and reference 22) follow a very similar structure, sharing text for introduction and data analysis. For instance, L410-430 are pretty similar to others in reference 22.
Therefore I will encourage the authors to modify the above mentioned in order to avoid what can be considered self-plagiarism. I do understand that both papers lies under a very similar scope, but you have to enhance descriptions, etc.
Considering the content of the paper, apart from the above, based on the CFP for the Special Issue the article can be include under the Remote laboratories for IoT topic, Platforms for IoT learning topic but specially on the Learning analytics topic.
In this sense, the paper could be accepted although in my personal opinion it doesn't offer a great advance in terms of novelty or design methodologies, as I said on my previous revision.
However, the data analysis of the impact of the new version is quite thoughrough and well explained. As said, the authors have some experience in this methodology as can be seen in reference 22.
Concerning the paper text, try to order references as it looks nicer when they are set consecutive as you read the document. It is clear that you have include some more references, but try to do it properly.
There are some typos and English errors, as for instance L368 (staring probably should be starting) and L289 on which should be rephrased. It seems that an extensive English review was done over the first version, but latter on some additions were made not being so careful.
Author Response

(The authors gave the same response as above.)

Round 3
Reviewer 1 Report
Most of my comments have been taken into account. The video viewing percentage is now correct in the text, but still swapped in Table 6.
For the opinion survey result (now Table 7), I had just given one example, but the problem applies to all statements. If you want to be precise, you cannot agree or disagree with “perceived usefulness”, “intention of use”, “easE of access” or “social influence” – you would have to say something like “… is perceived as very useful” or “… is perceived as completely useless” – with such a statement you can then agree or disagree.
I would still recommend the use of an English editing service to improve the language.
Author Response
See attached document.
Regards

Reviewer 2 Report
Previous issues have been addressed and Editors seem to have checked new content percentage.
Author Response
See attached document
Regards
